# Occurrence of Seahorses *Hippocampus* spp. in the Southernmost Part of Western Europe: A New Maximum Depth Record

**DOI:** 10.3390/ani14162328

**Published:** 2024-08-12

**Authors:** Ignacio Ruiz-Jarabo, Jorge Hernández-Urcera, Sira Pereira, Ignacio Sobrino, Juan A. López, Miquel Planas

**Affiliations:** 1Department of Marine Biology and Aquaculture, Institute of Marine Sciences of Andalusia—Spanish National Research Council (ICMAN-CSIC), 11519 Puerto Real, Spain; 2ECOBIOMAR Research Group, Instituto de Investigaciones Marinas—Spanish National Research Council (IIM-CSIC), 36208 Vigo, Spain; jurcera@iim.csic.es; 3Department of Marine Ecology and Resources, Instituto de Investigaciones Marinas—Spanish National Research Council (IIM-CSIC), 36208 Vigo, Spain; spereira@iim.csic.es; 4Centro Oceanográfico de Cádiz, Instituto Español de Oceanografía—Spanish National Research Council (IEO-CSIC), 11006 Cádiz, Spain; ignacio.sobrino@ieo.csic.es; 5Fundación Aula del Mar Mediterráneo, 29140 Málaga, Spain; presidencia@auladelmarmed.org

**Keywords:** conservation, citizen science, ecosystem services, Gulf of Cádiz, seahorse, *Hippocampus hippocampus*, *H. guttulatus*, Syngnathidae

## Abstract

**Simple Summary:**

Seahorses are iconic species facing population decline due to human action. Knowledge about seahorse distribution in Europe, particularly along the southwest coasts, remains scarce. This work has gathered information from diverse sources, including scientists, managers, professional fishermen, and the invaluable citizen-science platforms, resulting in novel information on the occurrence of two species of seahorses (*Hippocampus hippocampus* and *H. guttulatus*) in the Gulf of Cádiz and nearby areas. This study will allow for a more efficient management of local seahorse populations to safeguard their conservation.

**Abstract:**

Achieving sustainable resource use is a priority to meet future challenges. The Gulf of Cádiz, located in the Atlantic waters of southern Europe, is home to a significant fishing fleet due to the richness and diversity of its ecosystems. Managing this area is complex due to the diversity of variables, including social, ecological, and oceanographic factors. Therefore, multidisciplinary approaches are proposed for implementing conservation strategies. One strategy for defining area-based management measures is through the use of flagship species, such as seahorses. These emblematic animals can assist in defining such measures. However, there is currently scarce information on the occurrence of seahorses in the Gulf of Cádiz. In this study, we present the first occurrence data of two species of the genus *Hippocampus* (*H. hippocampus* and *H. guttulatus*) in this area. The Gulf of Cádiz is not only described as the southernmost region of their distribution in continental Europe, but it is also a significant landmark for the genus *Hippocampus*, as one adult was captured at a depth of up to 101 m. Five management areas based on differentiated benthic habitats are proposed. We believe that our study has the potential to significantly improve conservation of seahorses and induce a positive impact on the ecosystem.

## 1. Introduction

The conservation of marine habitats and biodiversity stands as a cornerstone for preserving the vitality and resilience of ocean ecosystems, which are pivotal for the overall health of our planet. Prioritizing sustainable fishing practices and marine conservation strategies is imperative in fostering the blue growth economy [1]. In pursuit of this goal, the implementation of area-based management measures emerges as a rational approach, facilitating the efficient utilization of resources [2]. Achieving effective marine conservation demands a holistic and integrated framework, underscoring the essential role of community engagement in fostering awareness and garnering support for sustainable initiatives. Despite progress, there remain substantial opportunities for enhancing fisheries management, notably in developed regions such as the Atlantic waters of southwestern Europe [3]. This underscores the ongoing need for continual improvement and innovation in marine conservation efforts to ensure the long-term conservation of our marine biodiversity.

The Gulf of Cádiz (ICES region IXa) is situated in the southwest of the Iberian Peninsula, constituting the southernmost marine ecosystems of continental Europe. It serves as a crucial nexus between the Atlantic Ocean and the Mediterranean Sea, facilitating significant exchanges between these two vast water bodies. Characterized by a wide continental shelf, the Gulf of Cádiz benefits from temperate waters and nutrient influx from multiple river basins [4,5,6].

Effective management strategies for marine resources must prioritize benthic habitats, given their provision of essential ecosystem services [7]. This imperative is particularly pronounced in the Gulf of Cádiz, an ecologically significant area where a diverse array of Atlantic and Mediterranean species converge [8]. The benthic megafaunal assemblages found here contribute significantly to overall biodiversity [9]. However, the Gulf also sustains a considerable fleet of bottom trawlers due to its high productivity [10]. Bottom trawling, known for its global adverse effects on marine biodiversity and ecosystem functioning, disrupts carbon turnover by mobilizing sediments, thereby impacting the abundance and biodiversity of meiofauna, as well as diminishing individual and overall biomass [11].

Overall, the ecological balance in the Gulf of Cádiz is threatened and an alarming loss of biodiversity has been described in recent decades due to the deterioration of marine ecosystems [12], although policy interventions are gradually ameliorating habitat deterioration [13]. Consequently, there is an urgent imperative to develop methodologies aimed at enhancing marine management while mitigating adverse impacts on the ecosystem. Simultaneously, sustainable approaches to fisheries production must be explored to ensure long-term ecological resilience and productivity.

Seahorses, alongside other syngnathids, have risen to prominence as iconic creatures serving as flagship species for ecosystem preservation [14]. However, seahorses face numerous threats that jeopardize their populations and survival. Their distinct biology and habitat preferences render them particularly vulnerable. Seahorses rely heavily on specific environments, and the degradation of these habitats due to coastal development, fishing, dredging, and pollution poses grave risks, depriving them of essential shelters, feeding grounds, and breeding sites [14]. Of significant concern is bottom trawling, a fishing technique involving the dragging of large nets along the seabed, which presents a formidable danger to seahorses. Incidentally captured in trawl nets, seahorses suffer elevated mortality rates as a consequence. These creatures coexist harmoniously with a diverse array of marine species, contributing to the overall biodiversity of their habitats [15]. Their presence enriches the ecosystem, fostering intricate networks of interactions among various organisms.

Efforts aimed at safeguarding seahorses extend beyond the protection of these unique beings; they encompass the preservation of the intricate balance of interactions they share with the diverse marine life inhabiting their habitats. By recognizing the pivotal role of seahorses in maintaining ecosystem health and diversity, conservation initiatives can effectively address the threats they face while fostering the resilience of marine ecosystems as a whole.

Genetic data revealed the presence of five species of seahorse in NE Atlantic, Mediterranean, and Black Sea waters: *Hippocampus algiricus*, *H. erectus*, *H. fuscus*, *H. hippocampus*, and *H. guttulatus* [16]. Only two of them are native to European waters, the short-snouted seahorse *H. hippocampus* (Linnaeus, 1758) and the long-snouted seahorse *H. guttulatus* (Cuvier, 1829). Both are listed as “Data Deficient” on the IUCN Red List and “Near Threatened” in the Mediterranean [17,18] because their populations are declining as a result of habitat degradation. They are also listed in Appendix II of the Convention on International Trade in Endangered Species of Wild Fauna and Flora (CITES) and in the Spanish List of Wild Species under Special Protection Status.

Seahorses commonly inhabit seagrass and macroalgae meadows, frequently associating with sponges and corals, predominantly in shallow, temperate waters [19] at depths not exceeding 50 m [20]. Distinctive biological traits of these species include low mobility and fertility, extended parental care, and mate fidelity [21]. These characteristics result in a scattered distribution and heightened vulnerability to habitat degradation. The decline in their natural populations is chiefly attributable to habitat degradation and incidental capture in fishing activities [14].

Conservation efforts targeting these species at the European level have emerged as a priority, with numerous scientific and stakeholder groups collaborating to address this imperative [22]. Nevertheless, despite the wealth of ecological and biological research conducted on these species [19,23,24], knowledge gaps persist regarding their geographical distribution. Closing these gaps is essential for informing more targeted conservation strategies and effectively safeguarding these vulnerable species and their habitats.

In the Gulf of Cádiz, seahorses belonging to the genus *Hippocampus* [25] and other syngnathids [26] have historically been documented, with close associations observed within seagrass meadows. However, a notable absence of seahorse sightings in the region has been documented in recent decades, likely stemming from a combination of factors including the trade of specimens for ornamental purposes and habitat degradation [21]. Regrettably, there have been no recent observations reported in the area, raising concerns regarding the population status and ecological conditions for these species [20]. This lack of recent data underscores the urgent need for renewed research efforts to assess the current status of seahorse populations in the Gulf of Cádiz, necessary to mitigate further declines and preserve their critical habitats.

The main aim of this study was to assess the occurrence of seahorses in the Gulf of Cádiz (Spain) and its surroundings. To accomplish this goal, we compiled data from available sources, including incidental captures documented by local artisanal fishermen, records from NGOs, Spanish national databases dedicated to bottom-trawling (https://digital.csic.es, accessed on 19 January 2024), and online platforms hosting citizen observations. The integration of these diverse data sets is expected to yield significant insights for resource managers, providing new perspectives on habitat conservation strategies within this exceptionally biodiverse marine area.

## 2. Materials and Methods

### 2.1. Ethics Statement

This study was partially conducted aboard research trawl and artisanal gillnet vessels. All seahorse captures were entirely incidental, and it is noteworthy that all captured animals were released alive back into the environment following capture. All procedures strictly adhered to the ethical standards outlined by the European Union guidelines (2010/63/UE) and complied with Spanish legislation (RD 53/2013) governing the protection of animals used for experimental and other scientific purposes.

### 2.2. Geographical Location, Vessels and Capture Characteristics

#### 2.2.1. Oceanographic Campaigns Aboard Bottom-Trawling Vessels

Seahorses for this study were obtained through hauls conducted during bottom-trawl surveys in Spanish waters off of the southwest Atlantic, specifically in the Gulf of Cádiz, from October 1992 to October 2023. All relevant data are accessible through the digital repository of the Spanish National Research Council (https://digital.csic.es, accessed on 19 January 2024). The hauls were conducted at depths ranging from 15 m to 750 m aboard three oceanographic vessels: O/V *Francisco de Paula Navarro* (length: 30.6 m; gross tonnage: 178 tonnes); O/V *Cornide de Saavedra* (length: 66.7 m; T.R.B.: 1150; engine power: 1651 kW); and O/V *Miguel Oliver* (length: 70 m; engine power: 2 × 1000 kW).

Sampling campaigns were carried out twice annually, in February–March and October–November, adhering to established international protocols [27]. The sampling gear utilized was equipped with a 44/60 BAKA trawl, towed at 3 knots for 1 h along a designated isobath (at a consistent depth) for each haul. To ensure uniform gear conditions, measurements of the net’s arrival and departure from the bottom, as well as its horizontal and vertical openings, were conducted using a MARPORT system (Milford, NH, USA). The start and end positions of each trawl were recorded using the Global Positioning System (GPS). Additionally, a conductivity–temperature–pressure (CTD) instrument was deployed in the net, while another probe continuously recorded surface water parameters. These campaigns also encompassed the collection of data on accompanying fauna and flora to establish relationships between seahorse occurrences and the sessile invertebrates present on the seabed. The captured animals were identified by trained scientific personnel and individually weighed with a resolution of 1 g.

#### 2.2.2. Artisanal Fisheries with Gillnets

The animals were captured using gillnets within the fishing grounds of the artisanal fleet of Chipiona, located in the fishing reserve at the mouth of the Guadalquivir River, within the Gulf of Cádiz. Vessel tracking via GPS was not conducted, and fishermen provided only approximate positions. The captures occurred between July 2022 and July 2023 at depths ranging from 10 m to 30 m, with specific depths for each catch remaining unknown. This study was conducted aboard commercial artisanal fishing vessels (<10 m in length), accommodating 2–3 persons and equipped with 5 L buckets of seawater collected from the sea surface. These buckets, thermally insulated and fitted with an aerator, maintained oxygen levels at saturation, aiding in the recovery of captured animals before their release. The nets were deployed before dawn, typically between 3:00 and 7:00 a.m., and retrieved after 1.5 to 6 h. The captured animals were identified by trained scientific personnel.

#### 2.2.3. Data Obtained from Open-Access Databases and Other Sources

Two of the records were obtained through personal communications. Researchers from the University of Cádiz (Spain) found two specimens in the Bay of Cádiz within the Gulf of Cádiz. Precise geographic coordinates were recorded for both sightings.

Scientific-grade seahorse sightings were integrated from the iNaturalist repository (https://www.inaturalist.org/) (accessed on 30 June 2024) [28], as documented in the GBIF database [29]. Thirty-three records were obtained from this platform. Notably, the geographic coordinates in this database are approximate.

The Aula del Mar Mediterráneo Foundation (FAMM) is a non-profit organization with a Spanish territorial scope (https://auladelmarmed.org/) (accessed on 30 June 2024). Its main objective is to design, promote, and develop projects for the conservation of biodiversity and natural heritage. Eighteen records were obtained from this NGO. Precise geographic coordinates were recorded for all sightings.

Seawatchers (OdM) is a citizen-science platform (https://www.seawatchers.net/) (accessed on 30 June 2024) coordinated by the Institute of Marine Sciences of Barcelona (CSIC). Experts from different national and international research institutions validate and comment on the observations received. Eight records were obtained from this platform. Precise geographic coordinates were recorded for all sightings.

### 2.3. Observation Map

The map with the geolocations of the observed animals was constructed using the software ArcGis 10.8. The accuracy of the geo-referenced captures/observations depended on the origin of the data. Once plotted, a geographically dependent pattern was observed, with five areas of seahorse captures in the Gulf of Cádiz and the Straits of Gibraltar.

### 2.4. Statistics

To explore potential differences in capture depth across observation areas for each species, a statistical analysis was conducted. Given that the data did not meet the assumptions of normality or homoscedasticity, a nonparametric Kruskal–Wallis test was applied, with area serving as the factor of variance. Statistical significance was considered at *p* < 0.05. Data are provided as mean ± SEM (standard error of the mean).

## 3. Results

A total of 120 seahorse specimens were identified, comprising 73 *H. hippocampus* and 47 *H. guttulatus*. Considering the various sources of information, the number of observations were as follows: 52 from scientific bottom-trawling surveys (IEO-CSIC), 6 from artisanal fisheries, 34 from the iNaturalist database, 8 from the ODM platform, 18 from FAMM ONG, and 2 from personal communications (Table 1; Figure 1).

### 3.1. Scientific Bottom-Trawling Surveys

The surveys conducted aboard oceanographic vessels from October 1992 to October 2023 yielded a total of 48 *H. hippocampus* and 4 *H. guttulatus* specimens (Appendix A). These oceanographic demersal trawl surveys covered the seafloor from 15 m to 750 m depth. Under these conditions, the minimum and maximum capture depths were 18 and 101 m, respectively, with a mean depth of 45 m and a statistical mode of 29 m (Appendix A, Figure 1). From the 57 fishing campaigns analyzed, seahorse specimens were observed in 12 autumn campaigns and 10 spring campaigns, accounting for 34.5% and 44.4% of all surveys, respectively. The average weight of the seahorses was 5 g, ranging from 1 to 18 g. According to FishBase (https://www.fishbase.us/, accessed on 19 January 2024 ) [30], all captured animals were classified as adults.

### 3.2. Artisanal Fisheries and Citizen Observations

The data on observations other than those from the scientific surveys are provided in Appendix A. The artisanal fishery operating at the mouth of the Guadalquivir River reported capturing 6 adult *H. hippocampus* between July 2022 and July 2023, at depths ranging from 10 to 30 m. The iNaturalist repository provided 14 and 20 observations of *H. hippocampus* and *H. guttulatus*, respectively, from shallow waters up to 8 m depth. Most *H. hippocampus* occurrences were reported in shallow waters (rocky or muddy bottoms) off Gibraltar (Bay of Algeciras, Mediterranean) and in the western part of the Gulf of Cádiz, washed up on beaches between Spain and Portugal. The majority of *H. guttulatus* specimens were observed near the Straits of Gibraltar in the Mediterranean. Citizen-science platform OdM data provided 3 *H. hippocampus* and 5 *H. guttulatus* occurrences around Algeciras Bay, whereas FAMM data provided 18 records of *H. guttulatus* in Algeciras Bay and its surroundings. Finally, 2 specimens of *H. hippocampus* were reported by personal observations in the Gulf of Cádiz.

### 3.3. Seahorse Distribution

Seahorse occurrences were distributed in five areas, as depicted in Figure 1:(i)Guadiana-Odiel rivers estuary, off the coast of Doñana National Park (A1);(ii)Guadalquivir River estuary (A2);(iii)Bay of Cádiz and its outer part, the area between the towns of Rota and San Fernando (A3);(iv)Southern area, off the coast of Conil (A4);(v)Bay of Algeciras, located outside the Gulf of Cádiz, in the Straits of Gibraltar (A5).

For *H. hippocampus*, statistically significant differences (*p* < 0.05, nonparametric Kruskal–Wallis test) were observed in the depth at which the seahorses were captured or observed based on the area of observation (Figure 2). The shallowest captures occurred at the mouth of Guadalquivir River (13.1 ± 2.2 m; n = 7 observations), followed by those in the Bay of Cádiz (24.3 ± 2.0 m; n = 27) and off the coast of Doñana (41.3 ± 19.5 m; n = 3), while the deepest seahorses were captured off the coast of Conil (58.2 ± 5.0 m; n = 21). Outside the Gulf of Cádiz, in the Bay of Algeciras, H. hippocampus was observed in shallow waters at 5.7 ± 1.2 m (n = 9).

*H. guttulatus* was observed only in three of these geographical areas: one seahorse was spotted near the mouth of the Odiel River, near Doñana (3 m depth); another was captured outside the Bay of Cádiz (26 m depth); three animals were captured off the coast of Conil (69.3 ± 8.3 m); and 43 seahorses were observed in shallow waters in Algeciras Bay, outside the Gulf of Cádiz (3.9 ± 0.3 m).

## 4. Discussion

Observations covering the last three decades in the Gulf of Cádiz, as a result of collaborative efforts of scientific campaigns and citizen science, have revealed the occurrence of two seahorse species, *H. hippocampus* and *H. guttulatus*, displaying a notable depth range extending from 1.5 to 101 m. Noteworthy was the groundbreaking capture of a specimen of *H. hippocampus* at 101 m, which is a new depth record for the genus *Hippocampus*, significantly surpassing the previously documented depth of 60 m for *H. reidi* [31].

The study revealed four primary areas of seahorse occurrence in the Gulf of Cádiz and one in the Bay of Algeciras, Strait of Gibraltar, each intricately linked to distinct benthic habitats:(i)Shallow bottoms dominated by macrophytes can be found along the coast of the Gulf of Cádiz (including the south of Portugal, the mouth of the Guadalquivir, and the inner sac of the Bay of Cádiz) and in the Bay of Algeciras [29,32];(ii)The demersal plain off Doñana is characterized by the recurrent presence of octocorals and ascidians [33];(iii)The deeper waters of the Bay of Cádiz harbour a significant number of hydrozoans and ascidians, as well as a well-developed community of deep-sea sponges and corals (results from the ARSA oceanographic surveys conducted by the IEO);(iv)Off the coast of Conil, the deepest depth at which seahorses were found, is dominated by rhodoliths, hexacorals, octocorals, and hydrozoans, as described during the ARSA surveys and other references [34].

This discovery accentuates the imperative for tailored conservation measures for these specific ecosystems.

Both seahorse species co-occur in the study area, being present in almost all the analysed zones but irregularly distributed. *Hippocampus hippocampus* was dominant in the Atlantic part of the study region, where *H. guttulatus* was poorly represented. Conversely, the latter was noticeably more abundant in the area comprising the Bay of Algeciras, Strait of Gibraltar area. Regarding the bathymetric distribution, the capture of specimens in the scientific bottom-trawling surveys at depths between 50 and 101 m was unexpected. These patterns could be attributed to interspecific ecological differences. Thus, while *H. guttulatus* is often associated with seagrass beds and commonly occupies shallow and patchier areas of high complexity, *H. hippocampus* is more frequently found in more open and deeper zones [35,36].

### 4.1. The Gulf of Cádiz

The effective management of the marine environment in the Gulf of Cádiz, as a distinctive geographical entity positioned between two continents (Europe and Africa) and two body waters (the Atlantic Ocean and Mediterranean Sea), is of paramount importance for future conservation and sustainability endeavors. The ocean currents coursing through the Straits of Gibraltar play a pivotal role in global current dynamics and exert control over ocean surface temperature [37,38,39].

The inflow of Atlantic water through the Strait of Gibraltar into the Mediterranean Sea carries sediment from southern Iberian Peninsula rivers, including the Guadalquivir, Guadiana, Tinto, and Odiel. This sedimentation process significantly contributes to the development of a remarkably diverse marine ecosystem [10]. Simultaneously, there is a notable outflow of high-salinity water from the Mediterranean towards the Atlantic. Regarding data from oceanographic surveys, the seahorse species documented in this study were previously unrecorded in the areas nearest to the Gulf of Cádiz, as evidenced by scientific reviews [20] and citizen-science platforms (e.g., iSeahorse or seawatchers.net). Nevertheless, technical reports from local associations corroborate this study’s findings in the Guadalquivir Estuary, especially in areas associated with gorgonians and coral banks, Cymodocea and Zostera meadows, hydrozoans, sponges, and pennatulaceans [33].

### 4.2. Seagrass Meadows

Seahorses and many other syngnathids typically inhabit seagrass and seaweed ecosystems [40]. These coastal habitats are of significant ecological importance, acknowledged as some of the most productive ecosystems worldwide. Thus, conservation efforts are warranted to safeguard their vital services [41].

The Bay of Cádiz is home to several macrophyte species, comprising three varieties of seagrass and numerous species of macroalgae [32]. Historically, the Bay has been associated with the occurrence of seahorses [25]; however, local extinctions have been declared by local fishermen due to illegal catches. Our data on seahorse occurrences both within and beyond Cádiz Bay suggest a potential recovery of the local populations. Nonetheless, to validate this hypothesis, systematic sampling focused on seahorse recruitment in the area would be crucial.

Both species of *Hippocampus* co-occur in Algeciras Bay, but *H. guttulatus* is clearly the dominant species. The Bay is a shallow area covered by seagrass and seaweed meadows [28]. Despite being situated outside the Gulf of Cádiz, there might be a genetic linkage between its seahorse populations and those inhabiting the Gulf of Cádiz. The detected genetic population subdivision in *H. hippocampus* underscores the necessity to manage populations as three separate units, with barriers to gene flow identified at Cape Finisterre and the Cape Verde frontal zone: English Channel + Bay of Biscay, Mediterranean Sea + Atlantic Iberian coast + Macaronesian Islands, and West Africa [42]. In the case of *H. guttulatus*, the species has the potential to form segregated populations, but genetic data indicate that the population east of the Strait of Gibraltar is part of the southern Iberian metapopulation (Atlantic coast) [43].

Hence, it is likely that seahorse populations in the studied area exhibit intraspecific genetic similarity, at least in *H. hippocampus*. However, it would be interesting to study migration rates and genetic flux in the area in a zone where the Atlantic Ocean and the Mediterranean Sea converge and to know the global genetic structure across large areas. An examination of the potential phylogenetic relationships between populations in the bays of Cádiz and Algeciras would yield significant value and be critical to effective population management. This exploration could foster cooperative management initiatives aimed at strengthening the conservation significance of both regions, given the presence of these iconic species.

### 4.3. Other Ecosystem Engineers

The exploration of rhodolith beds is gaining prominence due to their provision of valuable ecosystem services, fostering habitats conducive to supporting biodiversity comparable to coral reefs, seaweeds, and seagrasses [44]. Recent research has revealed a connection between the presence of rhodoliths and seahorses [45], suggesting potential ecological significance within seahorse habitats. The coastal region off Conil exhibits a diverse array of ecosystem-building organisms, fostering an environment that enhances biodiversity. This area encompasses patches of sandy, muddy, and rocky substrates. At depths ranging from 15 to 130 m, substrates are predominantly dominated by communities of sessile organisms such as gorgonians, sponges, hydrozoans, polychaetes, and coralline rhodolites of the genera *Litophyllum* and *Mesophyllum* [34], along with hexacorals and ascidians, as revealed in the present study.

The benthic richness in this area likely supports the presence of numerous specimens at depths ranging between 40 and 100 m. This is particularly noteworthy considering that European seahorse species have been observed in exceptional cases at maximum depths of 45–50 m in the marine shelf waters of the Mediterranean Sea [20]. The occurrence of *H. hippocampus* and, to a lesser extent, *H. guttulatus* in this study at depths between 50 and 101 m might be attributed to the abundance of resources and benthic faunal assemblages in the Gulf of Cádiz, encompassing crinoid beds, pennatulacean meadows, gorgonian fields, anthipatharian communities, and cold-water coral banks [8,9].

Thorough exploration of the Gulf of Cádiz, with special attention to physicochemical variables, is crucial. Since seahorses are visual predators, the amount of light in their environment is vital for their survival. Previous studies have indicated that the irradiance in the water column of the NW Mediterranean is similar to that of tropical seas [46]. Therefore, the Conil area, with the presence of rhodoliths in depth and water-column similarities to NW Mediterranean, may have an irradiance of around 5–15 µmol photons m^−2^ s^−1^ at 100 m depth [47]. The optimal culture irradiance for certain species of the *Hippocampus* genus ranges from 9 to 26 µmol photons m^−2^ s^−1^ [48,49]. Hence, the survival of seahorses should not be hindered by limitations in light availability. Assessing the prey availability in these demersal environments is also imperative. Despite the lack of existing data, these findings are exceptional for these seahorse species, warranting further comprehensive investigations.

### 4.4. Bottom Trawling

Trawling has well-documented disruptive effects on marine ecosystems [11], yet the fishing industry remains a cornerstone of societal and economic stability for many nations, including those bordering the Gulf of Cádiz [50]. Notably, within a context of resource scarcity, sustainable fisheries hold promise for profitability [51]. Hence, adopting an ecosystem-based approach to fisheries management requires the formulation of strategies and policy objectives rooted in practical operational goals [7].

The significant presence of a fishing fleet in the Gulf of Cádiz, comprising 126 trawlers predominantly targeting crustaceans and fish [52], poses a significant threat to seahorse populations. To address this, there is a growing call for increased consumer awareness and the exploration of sustainable alternatives to mitigate anthropogenic impacts on the seabed and the broader ecosystem, which includes seahorses [19]. Emphasizing a multidisciplinary approach is essential to identify vulnerable areas and implement effective conservation strategies [24].

Our observations highlight the critical need for information on accidental catches during fishing activities in deeper waters. Moreover, numerous fishing campaigns targeting other species can provide valuable insights into seahorse distribution. These surveys offer an invaluable means of data collection that is difficult to obtain through alternative methods. This comprehensive approach is indispensable for ensuring effective marine conservation and sustainable fisheries management in the Gulf of Cádiz.

### 4.5. Importance of Citizen Science Platforms

The scarcity of previous seahorse observations in the Gulf of Cádiz can be partially attributed to a lack of scientific observers or recreational divers, a phenomenon exacerbated by water turbidity caused by particulate matter from the Guadalquivir River. Notably, two seahorse (sub)populations have been documented on both sides of the Gulf of Cádiz, a large one in southern Portugal [23] and another in the westernmost part of the Mediterranean (the coast of Malaga), limited by the natural boundary of the Straits of Gibraltar [28]. Intriguingly, in both areas, a community of recreational divers actively contributes to providing information to scientists and society. However, it has been reported that seahorse surveys tend to focus on areas where seahorses are already known to be present or abundant, potentially constraining our ability to discern distribution and ecological patterns.

Most seahorse study surveys are conducted at moderate depths, occasionally exceeding 30 m. This limitation also applies to observations made by amateur or volunteer divers. Authors have noted that Atlantic seahorse populations favor coastal environments [20]. In contrast, while Mediterranean populations of *H. guttulatus* show a slight preference for confined shallow areas, *H. hippocampus*, characterized by a more marine disposition, tends to favor greater depths [53]. This underscores the importance of diverse and targeted survey efforts to uncover the nuanced ecological preferences of seahorse species in the Gulf of Cádiz.

### 4.6. Future Approaches

The presence of seahorse populations is confirmed along the entire southwestern coast of the Iberian Peninsula (Spain and Portugal) and in the Mediterranean Sea from the Strait of Gibraltar. The spatial connectivity of species with low mobility, such as seahorses, relies on the effective conservation of marine ecosystems [54]. Consequently, the establishment of marine protected areas in southern Europe is deemed essential to ensure the genetic variability of these species and secure their survival [24].

In the Gulf of Cádiz, active conservation strategies are in progress through the creation of areas with varying degrees of protection. This includes the Rio Tinto Protected Marine Reserve (established in 2015), the Guadalquivir Estuary Reserve (established in 2004 due to its significance for marine species recruitment), the Cádiz Bay Natural Park (established in 1989 due to its seagrass beds), and the Conil Fishing Reserve (currently being consolidated and rich in rhodolith beds). It is noteworthy that seahorses have been documented in recent years in these areas or in close proximity to them (Figure 1). The insights gained from this study can provide valuable assistance to managers in decision-making plans and contribute to the establishment of new protected areas grounded in scientific knowledge.

Future studies should prioritize examining the genetic distribution of these seahorse (sub)populations to assess their connectivity. The region experiences strong oceanic currents, with the predominant surface flow of water from the Atlantic to the Mediterranean, coupled with the deep countercurrent exiting the Mediterranean. The influence of ocean currents on shaping the distribution and prey availability for seahorses has been previously documented [55]. Within our study area, the dispersal and colonization of new habitats are significantly influenced by the distribution of species such as algae and planktonic organisms through these water flows [56]. Regardless of whether the seahorses of the Gulf of Cádiz share the same genetic population as those of Portugal and the Mediterranean, further exploration of their relationships is warranted. Additional research is crucial to delineate the population structure of the Gulf of Cádiz and identify specific points of conservation vulnerability.

Furthermore, collaboration with fishermen facilitates data collection through incidental catches of seahorses in their fishing operations. These unplanned captures provide essential information on seahorse occurrence, population dynamics, and potential threats, contributing to comprehensive ecological assessments. The active involvement of fishermen in research not only enhances scientific data but also promotes community participation in conservation efforts, establishing a symbiotic relationship between scientific inquiry and local expertise.

## 5. Conclusions

The distinctive oceanographic conditions of the Gulf of Cádiz underscore the necessity for comprehensive management strategies aimed at preserving its unique marine ecosystems. Our findings provide valuable insights for conservation efforts, supporting the establishment of marine protected areas and highlighting the importance of ongoing research to ensure the preservation of seahorses in this unique marine ecosystem.

To our knowledge, this study represents the first documentation of a *Hippocampus* species captured at depths beyond 100 m. This newfound information expands the distribution pattern of the two seahorse species inhabiting Europe and contributes to their enhanced ecological management. An intriguing avenue for exploration involves investigating whether individuals discovered at significant depths can act as colonization vectors to new areas, potentially influencing the level of population mixing.

Once again, community science, coupled with other data sources, has demonstrated its indispensable role in the management of marine resources, encompassing valuable flagship species and their habitats. This inclusive approach engages a diverse range of contributors and generates additional resources for conservation efforts. The collaborative efforts of community science are pivotal in advancing our understanding of marine ecosystems and strengthening conservation measures.

## Figures and Tables

**Figure 1 animals-14-02328-f001:**
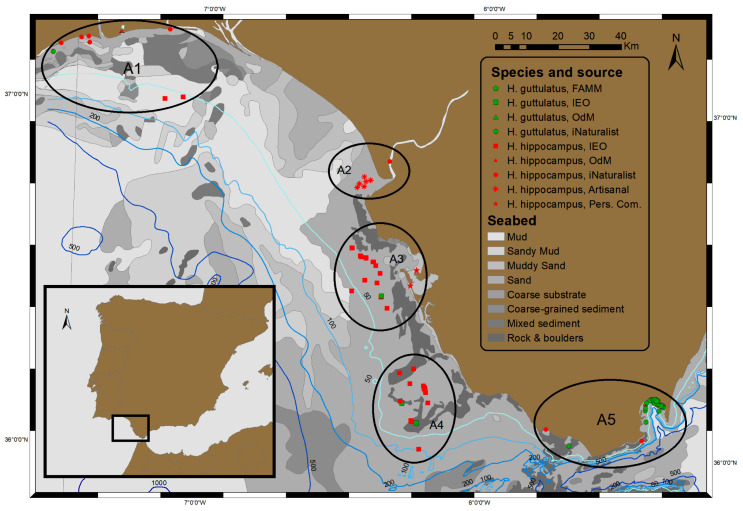
Sampling area off the southwest Atlantic in Spanish waters (Gulf of Cádiz) and locations of seahorses. The figure displays observations from different sources: squares represent trawl surveys, asterisks indicate observations of artisanal fisheries, circles and triangles denote citizen science data (iNaturalist and OdM, respectively), pentagons represent observations from FAMM, and stars are used for personal communications. The two species, *H. hippocampus* and *H. guttulatus*, are differentiated by red and green fillings, respectively. Ellipses indicate the 5 distribution areas identified (A1 to A5). The bathymetry is indicated by blue isobaths and the bottom type by a gray color code.

**Figure 2 animals-14-02328-f002:**
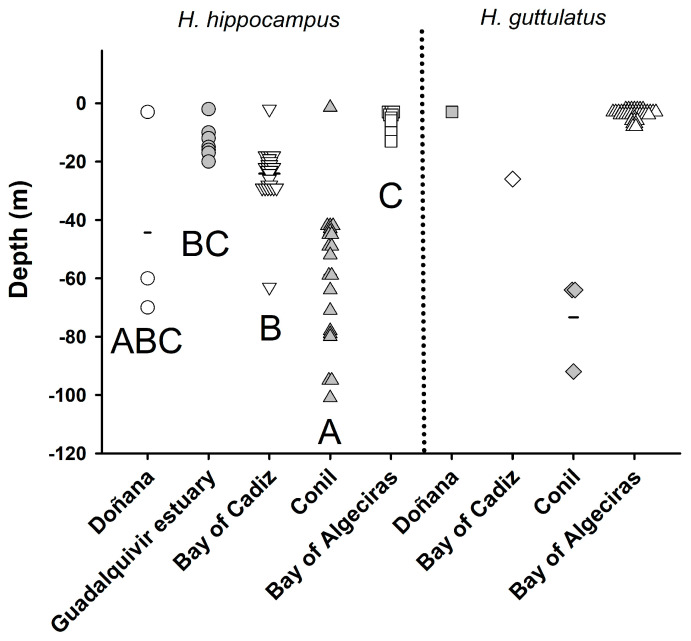
Density plot depicting the depth of seahorse captures in the Gulf of Cádiz and waters of the Bay of Algeciras, highlighting observations for each species (*H. hippocampus* or *H. guttulatus*). Horizontal lines denote the average depth for each zone. For *H. hippocampus*, significant differences between groups are denoted by different letters (Kruskal-Wallis test). Different symbols are used to show the animals sighted in each geographical area.

**Table 1 animals-14-02328-t001:** *H. hippocampus* and *H. guttulatus* observations provided by various sources.

Source	*H. hippocampus*	*H. guttulatus*	Total
Bottom-trawling surveys (IEO-CSIC)	48	4	52
Artisanal fisheries	6	0	6
iNaturalist	14	20	34
OdM	3	5	8
FAMM	0	18	18
Pers. communication	2	0	2
Total	**73**	**47**	**120**

## Data Availability

The raw data supporting the conclusions of this article will be made available by the authors on request.

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
