# Peer review of "Occurrence of Seahorses *Hippocampus* spp. in the Southernmost Part of Western Europe: A New Maximum Depth Record"

_animals, 2024, doi:10.3390/ani14162328_

Round 1

Reviewer 1 Report

Comments and Suggestions for Authors

The article is well written and provides important information for mapping priority areas for conservation that could be supported in the occurrence/distribution of seahorses in the Gulf of Cadiz and Baia Algiceras. Although, as the authors mention, more detailed studies are needed.

I only have two questions:

-You say that the minimum capture depth for H. hippocampus was 18m (line 227). Make it clear if the drags were only from 18 m depth.

-There really seems to be a different depth pattern for H. hippocampus (open and deeper waters) and H. guttulatus (shallow waters). How do you see the apparent coexistence of these species in Algeciras Bay (line 352)?

Author Response

Comments 1: You say that the minimum capture depth for H. hippocampus was 18m (line 227). Make it clear if the drags were only from 18 m depth.

Response 1: Modified as suggested. Now it reads as “These oceanographic demersal trawl surveys covered the seafloor from 15 m to 750 m depth. Under these conditions, the minimum and maximum capture depths were 18 and 101 m, respectively”.

Comments 2: There really seems to be a different depth pattern for H. hippocampus (open and deeper waters) and H. guttulatus (shallow waters). How do you see the apparent coexistence of these species in Algeciras Bay (line 352)?

Response 2: This is an interesting observation. It appears from the sightings in this study that both species occur in the same areas in both shallow and deep environments. However, H. hippocampus seems to be more abundant at greater depths (as commented in the second paragraph of the Discussion of this manuscript), with depth being a limiting element for the distribution of H. guttulatus. It would be very interesting to further explore what kind of differentiating elements, or interspecific ecological differences, occur between both species so that they can cohabit in the Bay of Algeciras. Future studies may be pointed towards the unravelling of their trophic positions and/or diets, or to the description of their preference for different anchoring substrates. These differences are certainly worth exploring.

Reviewer 2 Report

Comments and Suggestions for Authors

Main comment: this article is much too long and general. It should obviously be rewritten as short note (4 to 5 pages). Too many under-titles ans § are useless. Both make the paper so long and heavy!

- simple summery: useless, delete, at least 1 full latin name;

- abstract: idem; so general and obvious that the 1st half must be deleted; idem for introduction;

- what are conservation efforts for seahorse?

- fig 2: depth scale is usually going down; explain A B C.

Author Response

Comments 1: Main comment: this article is much too long and general. It should obviously be rewritten as short note (4 to 5 pages). Too many under-titles ans § are useless. Both make the paper so long and heavy!

Response 1: We are very sorry if the Reviewer feels that the manuscript is too long and heavy. However, although at some point we considered presenting it as a short note, we came to the joint conclusion (among all the authors) that the information obtained deserved a full article. We consider it necessary to make an effort to conserve these species in Europe, especially in those areas where there is no basic information about them (as is the case of the Gulf of Cadiz, in the south of the Iberian Peninsula). Therefore, this work, presented in full article format, can serve as a basis for future studies in the area. In a short note only the observation data of the animals could have been presented, greatly limiting the discussion of the results.  In the current version of the manuscript, in its discussion, a number of unique ecosystems present in the Gulf of Cadiz are named, as well as commercial activities carried out there (trawling), which are worth highlighting to facilitate the work of future studies in the area. We are grateful for the Reviewer's efforts, but unfortunately we feel that this manuscript deserves to be presented in full article format.

Comments 2: simple summery: useless, delete, at least 1 full latin name;

Response 2: The simple summary should be written according to the journal's guidelines. We cannot delete it. However, we have included full Latin names, as the Reviewer suggests. Thanks for this appreciation.

Comments 3: abstract: idem; so general and obvious that the 1st half must be deleted; idem for introduction;

Response 3: We are very grateful for the Reviewer's efforts, and appreciate her/his comments, but prefer to leave the summary and introduction as they are. We believe that the information presented in these sections is useful and significant.

Comments 4: what are conservation efforts for seahorse?

Response 4: Seahorse conservation requires a multi-faceted approach addressing threats such as habitat loss, overfishing, and trade. Current conservation efforts in Spain primarily concentrate on establishing the geographic distribution and habitat characteristics of seahorse populations. To achieve this, widespread public awareness campaigns have been initiated, engaging citizens in data collection and ecosystem stewardship. Effective dissemination of findings to marine conservation stakeholders is ongoing. It is anticipated that the combined efforts of these initiatives and the Hippo-DEC project will catalyze the implementation of robust government protections for local seahorse populations. Among other measures, establishing or expanding MPAs, implementing fishing regulations, improving monitoring and increasing awareness are also other conservation actions to consider.

Comments 5: fig 2: depth scale is usually going down; explain A B C.

Response 5: We have modified the figure as suggested by the Reviewer. The letters are explained in the figure caption: “For H. hippocampus, significant differences between groups are denoted by different letters (Kruskal-Wallis test)”.

Reviewer 3 Report

Comments and Suggestions for Authors

This is an interesting, relevant, and well-designed study that provides more information on seahorse occurrence in the Gulf of Cadiz area. Valid information on seahorse distribution along their known geographic range is fundamental for mapping the distribution of these increasingly threatened species, and this study provides that for a specific area. The manuscript is also well-written, so it fully merits publication as is.

Author Response

Comments 1: This is an interesting, relevant, and well-designed study that provides more information on seahorse occurrence in the Gulf of Cadiz area. Valid information on seahorse distribution along their known geographic range is fundamental for mapping the distribution of these increasingly threatened species, and this study provides that for a specific area. The manuscript is also well-written, so it fully merits publication as is.

Response 1: We would like to thank the Reviewer for these words.